# Andrographolide Induces G2/M Cell Cycle Arrest and Apoptosis in Human Glioblastoma DBTRG-05MG Cell Line via ERK1/2 /c-Myc/p53 Signaling Pathway

**DOI:** 10.3390/molecules27196686

**Published:** 2022-10-08

**Authors:** Nurul Syamimi Othman, Daruliza Kernain Mohd Azman

**Affiliations:** Institute for Research in Molecular Medicine (INFORMM), Universiti Sains Malaysia, Gelugor 11700, Malaysia

**Keywords:** andrographolide, DBTRG-05MG, WST-1 assay, scratch assay, cell cycle, apoptosis, c-Myc, p-53, co-immunoprecipitation, qRT-PCR

## Abstract

Human glioblastoma multiforme (GBM) is one of the most malignant brain tumors, with a high mortality rate worldwide. Conventional GBM treatment is now challenged by the presence of the blood–brain barrier (BBB), drug resistance, and post-treatment adverse effects. Hence, developing bioactive compounds isolated from plant species and identifying molecular pathways in facilitating effective treatment has become crucial in GBM. Based on pharmacodynamic studies, andrographolide has sparked the interest of cancer researchers, who believe it may alleviate difficulties in GBM therapy; however, it still requires further study. Andrographolide is a bicyclic diterpene lactone derived from *Andrographis paniculata* (Burm.f.) Wallich ex Nees that has anticancer properties in various cancer cell lines. The present study aimed to evaluate andrographolide’s anticancer effectiveness and potential molecular pathways using a DBTRG-05MG cell line. The antiproliferative activity of andrographolide was determined using the WST-1 assay, while scratch assay and clonogenic assay were used to evaluate andrographolide’s effectiveness against the cancer cell line by examining cell migration and colony formation. Flowcytometry was also used to examine the apoptosis and cell cycle arrest induced by andrographolide. The mRNA and protein expression level involved in the ERK1/2/c-Myc/p53 signaling pathway was then assessed using qRT-PCR and Western blot. The protein–protein interaction between c-Myc and p53 was determined by a reciprocal experiment of the co-immunoprecipitation (co-IP) using DBTRG-05MG total cell lysate. Andrographolide significantly reduced the viability of DBTRG-05MG cell lines in a concentration- and time-dependent manner. In addition, scratch and clonogenic assays confirmed the effectiveness of andrographolide in reducing cell migration and colony formation of DBTRG-05MG, respectively. Andrographolide also promoted cell cycle arrest in the G2/M phase, followed by apoptosis in the DBTRG-05MG cell line, by inducing ERK1/2, c-Myc, and p53 expression at the mRNA level. Western blot results demonstrated that c-Myc overexpression also increased the production of the anti-apoptotic protein p53. Our findings revealed that c-Myc and p53 positively interact in triggering the apoptotic signaling pathway. This study successfully discovered the involvement of ERK1/2/c-Myc/p53 in the suppression of the DBTRG-05MG cell line via cell cycle arrest followed by the apoptosis signaling pathway following andrographolide treatment.

## 1. Introduction

Glioblastoma multiforme (GBM) is the deadliest and most difficult to treat brain cancer disease. Patients diagnosed with GBM have a one-year survival rate because of the limitations of current treatments, which have unfavorable side effects on patients and make it difficult to remove the tumor from a sensitive area of the brain [1]. In addition, to achieve the Sustainable Development Goal (SDG) objective 3.4.1, which aims to minimize early death from noncommunicable diseases (NCDs) by one-third by 2030 and improve mental health and wellbeing over all ages, particularly the mortality rate related to cancer must be improved [2]. Current treatment or alternative therapies for brain cancer were intended. As a result, researchers started to find natural products that could be developed as small-molecule targeted therapies for targeting cancer cells via different mechanisms of action for GBM treatment. Andrographolide is a bioactive compound in *Andrographis paniculata* and is widely used as a traditional medicine in Malaysia, Indonesia, China, and India to treat diabetes and malaria [3]. Andrographolide is the lactone diterpenoid with a colorless crystalline solid with a bitter taste, which exhibits biological activities including antibacterial, antivirus, anti-inflammatory, anti-diabetic, and anticancer. The chemical structure name of andrographolide is 3α,14,15,18-tetrahydroxy-5ß,9ßH,10α-labda-8(20),12-dien-16-oic acid γ-lactone, and the molecular formula and weight are C_20_H_30_O_5_ and 350.4 g/mol [4].

More interestingly, andrographolide is a potential drug candidate for the treatment of any diseases related to the central nervous system (CNS) because it has a neuroprotective effect in the CNS by preserving the neuronal structure and function [5]. It is also a polar compound with low molecular weight that is able to penetrate the phospholipid of membranes of the blood–brain barrier (BBB) by passive diffusion [6]. Hence, many researchers started to focus on andrographolide as the potential small-molecule drug that can be used in brain cancer treatment.

According to the literature, andrographolide can be developed further as an anticancer drug to treat cancer cells through various mechanisms such as apoptosis, autophagy, cell adhesion control, proliferation, migration, invasion, anti-angiogenic activity, and other miscellaneous acts [7]. Further investigations have shown andrographolide’s potential in suppressing the growth of various cancers, including human melanoma [8], colon cancer [9], and human prostate cancer [10], by inducing cell cycle arrest and apoptosis, such as in ROS/JNK, ERK-p53 and p13/AKT/p53 signaling pathways [11]. However, the mechanical action of andrographolide on human DBTRG-05MG cell lines to inhibit cell growth by induction of cell cycle arrest and apoptosis is yet to be understood in detail. Additionally, a previous study reported that andrographolide could significantly interact with the MAPK/ERK1/2 pathway involved in cell death and migration [12].

The C-myelocytomasis (c-Myc) oncogene is a transcriptionally encoded gene that regulates the expression of over 100 target genes involved in apoptosis, cell cycle arrest, cell growth, survival, and differentiation [13]. C-Myc is a well-known protein that can induce tumorigenesis when upregulated in many cancers such as colon cancer [14], breast cancer [15], liver cancer [16], and head and neck squamous cell carcinoma (HNSCC) [17]. Previous studies have also mentioned that the overexpression of c-Myc activates many downstream genes, such as p53, gadd45, cdc25A, Cyclin A, Cyclin D, and CDK4, which trigger cell cycle arrest and apoptosis [18]. Therefore, this study aimed to use a specific compound, andrographolide, in GBM as the primary source of the drug by targeting the upstream and downstream regulators of the c-Myc signaling pathway such as MAPK/ERK1/2, Wnt/β-catenin, Hedgehog, Hippo, p53, NF-Κb, STAT, and p13-K/AKT/ERK [19]. Then, this study aimed to investigate whether c-Myc or c-Myc’s target genes can be targeted by andrographolide for cancer therapy. In addition, the common pathway involved in activating c-Myc in GBM is discussed. 

Moreover, the development of a new drug that can directly target c-Myc still needs to be explored because many researchers have attempted to explain the molecular mechanism of a new drug involved in activating c-Myc in human cancer. After all, information and knowledge regarding the expression and possible interaction of c-Myc in cancer remain insufficient. This review aims to provide information and knowledge related to c-Myc in the development of GBM. This study also aimed to determine the interaction between andrographolide and c-Myc, as well as how andrographolide is involved in upstream and downstream c-Myc signaling pathways in the human brain cell line, as there is no evidence of this having been yet studied. Then, this study aimed to investigate whether c-Myc or c-Myc’s target genes can be targeted by andrographolide for cancer therapy.

## 2. Results 

### 2.1. Andrographolide Suppressed Cell Viability of DBTRG-05MG and SVGp12 Cell Line

Human brain cancer cell lines (DBTRG-05MG) and normal brain cell lines (SVGp12) were exposed to nine different concentrations of andrographolide (0.781–200 µM) for 24 h, 48 h, and 72 h (Figure 1A,B). The WST-1 assay was conducted to determine the cell viability of andrographolide towards DBTRG-05MG cells. The cell viability of DBTRG-05MG cells was found to be time- and concentration-dependent, as the cell viability decreased while andrographolide concentration increased. Based on Table 1A, the lowest LC_50_ values were determined at 13.95 µM for 72 h, which demonstrated the greatest cytotoxic effects and were chosen as the main active dose and time in the subsequent experiments. The LC_50_ values for the normal cell line cannot be computed because, as shown in Figure 1B, normal cell viability was maintained ≥90% after being treated with the final concentrations (200 μM) of andrographolide for 24 h. This indicates that andrographolide has anticancer activity with a specific toxicity effect appearing to be more significant in cancer cell lines than in normal cell lines. This cell study employed temozolomide to treat DBTRG-05MG as a positive control to show the accuracy of the assay. Temozolomide is the current chemotherapy drug used to treat GBM disease. This experiment shows that andrographolide completely inhibits cell viability of cancer cell lines at low concentrations (Figure 1C), with the recorded LC_50_ values below 5.8 μM at 24 h of incubation (Table 1B). 

### 2.2. Scratch 

#### Scratch Assay

The effect of andrographolide on cell migration changes of the DBTRG-05MG cell line was assessed using the scratch assay. Figure 2A demonstrates that a smaller number of cells migrated into the middle wound area throughout the time of treatment with andrographolide compared to the control. This reveals that andrographolide (concentration: 13.95 µm) delayed healing, decreased migration ability, and did not improve the wound closure in the DBTRG-05MG cell line after 72 h of incubation compared to the control. From microscopy imaging, the DBTRG-05MG cell line with andrographolide showed minimal cell proliferation activity in the wound area compared to the control group. The percentages of wound closure area after 24, 48, and 72 h were significantly higher when the cell line was treated with andrographolide compared to the control group (Figure 2B). These findings show that andrographolide delayed the migration of the DBTRG-05MG cell line in a time-dependent manner, since the number of cells migrating decreased from 24 to 72 h.

### 2.3. Clonogenic Assay

To confirm the effect of andrographolide in reducing cell viability of DBTRG-05MG cell lines, the colony formation of cell lines treated with 13.95 µM (LC_50_ values) and 27.9 µM (2LC_50_ values) of andrographolide at 72 h was observed. Figure 3A shows the number of colonies that survived after treatment was reduced with the increasing concentration after 72 h compared to the control. From Figure 3B, it can be seen that the percentage of the surviving fraction was significantly reduced from 40% to 6% at 13.95 μM to 27.9 μM of andrographolide, respectively. These results revealed that andrographolide reduces the survival of DBTRG-05MG cell lines in a concentration-dependent manner compared to the control.

### 2.4. Andrographolide Induces G2/M Cell Cycle Arrest Followed by Apoptosis in DBTRG-05MG Cancer Cell Lines

Cell cycle arrest was determined by staining the DNA of apoptotic DBTRG-05MG cells with propidium iodide after treatment with different andrographolide concentrations (13.95 μM and 27.9 μM) compared to the control cell. Figure 4A represented the cell cycle distribution of DBTRG-05MG cell lines at G0/G1, S, and G2/M phases after being treated with andrographolide. This finding shows that as the concentration of andrographolide increases, the cell begins to accumulate in the G2/M phase. Figure 4B also shows andrographolide significantly increases the percentage of cells in the G2/M phase compared to control (1.89%), LC_50_ (8.60%), and 2LC_50_ (8.95%). This study reveals that andrographolide induced cytotoxic effects through G2/M phase arrest in a concentration-dependent manner.

The apoptosis analysis was elucidated through the flow cytometry technique using double staining with annexin V-FITC and PI. The scattered diagram was obtained from flow cytometry analysis of cancer cell lines after exposure to different andrographolide concentrations (13.95 μM and 27.9 μM) compared with the control cell (Figure 4C). There was a concentration-dependent manner in the apoptotic cell accumulated in Q2 and Q4. Q2 refers to late apoptotic cells, Q4 refers to early apoptotic cells, and Q1 and Q3 quadrants refer to necrosis cells and viable cells. The percentage of apoptotic cells in control cells was 0.01%, and after the cell lines were exposed to 13.95 μM and 27.9 μM of andrographolide for 72 h, the percentage of the apoptotic cell was 5.2% and 16.5%, respectively (Figure 4D).

### 2.5. The mRNA and Protein Expression Level of ERK1/2, p53, and c-Myc Was Increased in DBTRG-05MG Treated with Andrographolide

The effect of andrographolide on the relative expression of ERK1/2, p53, and c-Myc in DBTRG-05MG cells was evaluated by Quantitative Real-Time Polymerase Chain Reaction analysis (qRT-PCR). In Figure 5A, the results show that the relative expression of c-Myc, ERK1/2, and p53 in DBTRG-05MG significantly increased in a concentration-dependent manner after andrographolide treatment. When DBTRG-05MG was treated with low (LC_50_: 13.95 μM) and high (2LC_50_: 27.21 μM) concentrations of andrographolide, the relative expression of c-Myc significantly increased from 1.12-fold to 5.73-fold. The relative expression of ERK1/2 and p53 were also increased from 1.72-fold to 2.87-fold and from 1.48-fold to 2.25-fold, respectively, compared to untreated DBTRG-05MG cells. This finding was further analyzed via Western blot to determine the protein expression level of c-Myc, ERK1/2, and p53. Figure 5B shows the intensities of bands that denote the protein levels of c-Myc and p53 that increased in a dose-dependent manner after DBTRG-05MG cells were treated with andrographolide. The relative intensities of c-Myc (Figure 5C) and (Figure 5D) p53 normalized to α-tubulin for untreated cells and treated cells with low (LC_50_: 13.95 μM) and high (2LC_50_: 27.21 μM) concentrations of andrographolide were increased, respectively. The upregulation of c-Myc markedly increased the expression of p53. Furthermore, c-Myc was implied to interact with p53 constitutively, and this interaction can be evaluated through co-IP.

### 2.6. The Protein–Protein Interaction of c-Myc and p53 Involved in DBTRG-05MG Treated with Andrographolide

The study on the interaction or relation between c-Myc and p53 protein, co-IP, and Western blot analysis was performed accordingly. The reciprocal co-IP assay was performed with the total protein lysate from the DBTGRG-05MG after exposure to the andrographolide. Figure 6A and B show the co-IP result of p53 and c-Myc from DBTRG-05MG total cell lysate treatment with andrographolide. The p53 and c-Myc proteins were precipitated for 1 h at 4 °C using protein G Sepharose beads and specific antibodies (IP). Precipitated proteins were separated by SDS-PAGE and immunoblotted with p53 and c-Myc antibodies (IB). Figure 6A is the lane immunoprecipitated (IP) result for p53 and probed with anti-c-Myc in the Western blot analysis, which showed the presence of the band for c-Myc protein (62 kDA). Figure 6B is the lane immunoprecipitated (IP) result for c-Myc and probed with anti-p53 in the Western blot analysis, which showed the presence of the band for p-53 protein (50-44 kDA). The presence of a band in the lane IP in Figure 6A,B shows that c-Myc and p53 interacted with each other or bound together to induce the apoptosis signaling pathway. The interaction of c-Myc and p53 in DBTRG-05MG after being treated with andrographolide was successfully determined in this study. Protein G Sepharose beads mixed with total cell lysate without any antibody were used as a negative control to observe the unspecific binding of the beads.

## 3. Discussion

The current anticancer drugs in conventional treatments are becoming resistant to cancer, particularly in GBM, which is one of the uncommon brain cancers produced from glial cells of the central nervous system (CNS). GBM is also the world’s leading cause of death due to the limitations of current conventional therapies such as surgery, radiation, and chemotherapy [20]. This has led to the development of andrographolide as a potential small-molecule drug in GBM treatment by targeting different molecular pathways. Andrographolide has intrigued many researchers because of its ability to cross the BBB and inhibit other cancer cell lines [5]. This study aimed to provide information and knowledge on the cytotoxic effect of andrographolide on DBTRG-05MG cell lines, as well as to understand the mechanism of cell death that occurred when DBTRG-05MG cell lines were treated with andrographolide.

In this study, the andrographolide’s effectiveness in DBTRG-05MG cell lines was assessed by a different assay that covers cytotoxicity, proliferation, and the molecular mechanism of cell death. The cell cytotoxicity and proliferation of DBTRG-05MG were successfully determined by WST-1, scratch, and clonogenic assays. This present study found that andrographolide significantly suppressed the cell viability of the DBTRG-05MG cell line in a time- and concentration-dependent manner. The cytotoxicity of DBTRG-05MG cell lines towards andrographolides was higher compared to SVGp12 normal cell lines with an LC_50_ of 13.95 μM at 72 h. This finding can be supported by a previous study that tested andrographolide using various cancer cell lines with LC_50_ values ranging from 2.25 to 44.25 µM [21]. Hence, the LC_50_ value 13.95 μM, obtained from this study, was in the acceptable range reported by the previous study. This result also indicated that andrographolide showed more cytotoxicity toward cancer cell lines than a normal cell line. However, these studies remain controversial because no previous study was reported on andrographolide’s effectiveness in reducing the cell viability of the DBTRG-05MG cell line. This is because many studies used other types of cell lines to mimic the GBM diseases, such as Uppsala 87 Malignant Glioma (U-87 MG) [22], GBM8401, and U251 cells [23], which significantly support our findings that andrographolide has shown anticancer properties in treating GBM disease via cell viability reduction.

Then, this study was continued with the clonogenic assay, which showed that andrographolide was able to inhibit the cell proliferation of DBTRG-05MG in a concentration-dependent manner by reducing the colony formation of cell lines over time. A few studies were conducted to evaluate the effect of andrographolide on the colony formation of human glioblastoma cell lines that involved a different type of cell line, such as U-87 MG [22]. The previous study evidently inhibited the GBM colony formation in a concentration-dependent manner. However, no study was conducted on the effect of colony formation toward the DBTRG-05MG cell line after being treated with andrographolide. Hence, the present study was successful in providing the information that andrographolide exhibited cytotoxicity by reducing the colony formation after a 72 h treatment. Additionally, the scratch assay was performed on DBTRG-05MG cells and revealed that andrographolide delayed the cell migration into the scratch wound area in a time-dependent manner. A previous study also mentioned the ability of andrographolide to delay the migration in GBM cells [23]. These reports support our research findings because DBTRG-05MG cell lines are very slow in proliferation and reduce the cell migration activity in the middle of the wound area after being exposed to andrographolide at 24, 48, and 72 h compared to the control group. Therefore, our results suggested that andrographolide has the potential to induce or prevent the cell migration of the DBTRG-05MG cell line. 

This study also revealed that andrographolide induced a cytotoxic effect on the DBTRG-05MG cell line through G2/M cell cycle phase arrest followed by apoptosis in a concentration-dependent manner. The percentage of cells arrested at the G2/M phase in DBTRG-05MG cells after treatment of andrographolide at 13.95 μM for 72 h increased significantly compared with the control. Then, the apoptotic cell increased in the DBTRG-05MG cell line after andrographolide of 13.95 μM treatment for 72 h. According to our findings, andrographolide successfully suppressed the cell proliferation of the DBTRG-05MG cell line and was arrested at the G2/M phase, followed by apoptosis. Then, to gain information on the molecular mechanism involved in the cell cycle arrest and cell apoptosis after the DBTRG-05MG cell line was treated with the andrographolide, the ERK1/2/c-Myc/p53 signaling pathway was investigated in this study. This study was successful in detecting the mRNA and protein expression levels of c-Myc, ERK1/2, and p53 genes via qRT-PCR and Western blot analysis. Our finding successfully found that the upstream and downstream regulator genes of the c-Myc signaling pathway involved ERK1/2 and p53. 

The mRNA and protein expression level of ERK1/2 increased accordingly after andrographolide treatment compared with control. This finding was consistent with the previous finding that reported andrographolide can interact and activates ERK1/2 [24]. The previous study also proved that andrographolide could activate ERK1/2- and p53-induced apoptosis in C6 glioma cells [25]. This study also was extended to determine the level of c-Myc expression, and it was discovered that c-Myc was overexpressed in DBTRG-05MG following andrographolide treatment. This result indicated that ERK1/2 activation was related to c-Myc overexpression in DBTRG-05MG cells because previous studies have shown that ERK1/2 is the upstream regulator for the c-Myc transcription factor, which regulates many downstream genes involved in cell proliferation, apoptosis, and transcription [26]. In addition, andrographolide treatment inhibits DBTRG-05MG cell growth, which induces apoptosis because the pro-apoptotic protein p53 is activated, as demonstrated by the elevation of p53 at mRNA and protein expression levels. This discovery explained how post-translational and post-transcriptional mechanisms affected the protein and mRNA levels in DBTRG-05MG, because ERK1/2, c-Myc, and p53 mRNA expression levels were consistent with their protein expression levels after treatment with different concentrations of andrographolide.

Our findings showed that the expression of ERK1/2 was upregulated in DBTRG-05MG along with the upregulation of the c-Myc and p53 after andrographolide treatment. However, another study reported that andrographolide treatment suppressed lung cancer cell growth by downregulating c-Myc expression [27]. Our findings contradicted the previous studies because in the DBTRG-05MG brain cancer cell line after andrographolide treatment, c-Myc was overexpressed to inhibit the cancer cell, as can be supported well with our WST-1, scratch, and clonogenic assays. This study suggested that ERK1/2 activated c-Myc transcription, which can eventually create a wide range of proteins across the DBTRG-05MG cell, including p53, which can cause apoptotic cell death in a brain cancer cell line. Moreover, researchers have started to discover the unique function of c-Myc on gene and protein expression in various cancer cells. The previous study found that increasing the expression of c-Myc leads to increased transcription elongation by RNA polymerase II (RNA Pol II) and higher transcript levels per cell. Then, the transcriptional amplification occurs due to high levels of c-Myc in cancer cells, producing elevated levels of transcripts from the existing gene expression program of tumor cells [28]. A recent study highlighted that the c-Myc transcription factor has a dual role in tumor cells because it can activate and repress the various downstream pathways that can induce proliferation or apoptosis [29]. Many researchers implied that the influence of c-Myc on cell apoptosis was very complex and remained controversial. As a result, this study discovered that elevated levels of c-Myc in cancer cells result in newly activated “Myc target genes”, such as p53, for cell cycle arrest at the G2/M phase followed by apoptosis to remove undesirable or abnormal cells before continuing the cell cycle progression. More interestingly, earlier research has revealed that p53 influences the cell cycle checkpoint during G2/M, and cell arrest occurs before mitosis [30], which is supported by our findings via flow cytometry analysis. 

Our findings were extended with a co-IP analysis to assess the interaction of c-Myc with p53 via this signaling pathway. After being treated with andrographolide, c-Myc and p53 were able to form a protein complex in DBTRG-05MG; both proteins were bound together to initiate an apoptotic signaling pathway. The positive interaction between c-Myc and p53 also was discovered in the nasopharyngeal carcinoma (NPC) cell line for apoptosis [31]. Then, a similar finding was reported on the interaction of c-Myc and p53 in a colorectal cancer cell, which suggested c-Myc binding at the promoter of p53 and that c-Myc upregulated p53 to induce the apoptosis response [32]. This finding remains controversial because this study is the first to demonstrate the interaction between c-Myc and p53 protein that worked or bound together to mediate apoptosis in the human brain DBTRG-05MG cell line. 

Based on co-IP results, c-Myc displayed a single band (Figure 6A), whereas p53 exhibited two closely spaced bands migrated broadly at 50-44 kDa (Figure 6B) in the Western blot analysis using the anti-p53 antibody (DO-1) from Santa Cruz. This implies that DBTRG-05MG cells express various p53 isoforms, which was discovered by previous studies that showed the anti-p53 antibody (DO-1) could identify transactivation domain p53 isoforms such as p53α, p53β, and p53γ [33,34]. The previous study also successfully identified the p53 isoform in GBM that migrates with apparent molecular weights within this range: Δ40p53 (48 kDa); p53β (46 kDa); p53 γ (46 kDa), and Δp53 (44 kDa) [35]. This finding suggested that the 44 kDa band was Δp53. This discovery provides fresh knowledge and opens the door to additional functionalities of p53 isoforms.

Overall, evidence suggests that andrographolide at low concentrations induced cell cycle arrest and apoptosis via the c-Myc signaling pathway. This finding suggested that ERK1/2 acts as an upstream regulator, enhancing the activation of the c-Myc signaling pathway by modulating the activity of apoptotic proteins such as p53. Figure 7 depicts an overall schematic presentation of the proposed mechanism of andrographolide’s anticancer impact on DBTRG-05MG cells, which highlights an ERK1/2/c-Myc/p53 signaling pathway that triggered apoptosis. This discovery has provided new insight into the ERK1/2/c-Myc/p53 signaling pathway in brain cancer, which may include one of the treatment options.

## 4. Materials and Methods

### 4.1. Materials

#### 4.1.1. Chemical Reagents

Andrographolide, temozolomide (anticancer drug), and crystal violet were supplied by Sigma–Aldrich (USA). Andrographolide and temozolomide were dissolved in DMSO and kept at −20°C until needed. RPMI 1640, DMEM, trypsin-EDTA, and FBS were purchased from Gibco (USA). Penicillin–streptomycin was purchased from Nacalai Tesque (USA). WST-1 (sodium 5-(2, 4-disulfophenyl)-2-(4-iodophenyl)-3-(4-nitrophenyl)- 2H-tetrazolium inner salt) was purchased from Roche (Germany). The PI/RNase staining buffer and Annexin V-FITC Apoptosis Detection Kit were obtained from BD Pharmingen (USA) and Invitrogen (Austria).

#### 4.1.2. Human Cell Lines

Human glioblastoma cell lines DBTRG-05MG and normal glial cell lines SVGp12 were supplied by the American Type Culture Collection (ATCC), USA. Both cell lines were used in this study and were cultured in complete RPMI and DMEM supplemented with 10% (*v*/*v*) Fetal Bovine Serum (Thermo Scientific, Waltham, MA, USA) and 1% penicillin–streptomycin (Nacalai Tesque, Kyoto, Japan). The cells were maintained in standard cell culture conditions, a humidified incubator with 5% CO_2_ at 37 °C.

### 4.2. Methods

#### 4.2.1. Cytotoxic Activity Assay

The cytotoxic effect of andrographolide was assessed by WST-1 assay in DBTRG-05MG and SVGp12 cell lines. Both cell lines were seeded into 96-well plates at 6 × 10^3^ cells/well in 100 µL of complete medium growth. The cells were incubated at 5% CO_2_ at 37 °C overnight or until the cells reached 70–80% confluently. Then, the cells were treated for 24, 48, and 72 h with varying concentrations of andrographolide (0.781–200 µM), with the final concentration of DMSO adjusted to 0.2% for all concentrations of andrographolide. After incubation, the old medium was discarded, and the cell was washed using PBS. Then, 100 µL of the sterile medium of complete RPMI was supplemented with 10% (*v*/*v*) Fetal Bovine and 1% penicillin–streptomycin, and each well-received 10 µL of cell proliferation reagent WST-1 (Roche, Mannheim, Germany). The plates were incubated in 5% CO_2_ at 37 °C for 1 h, and the absorbance (Abs.) was measured using the ELISA reader (Multiscan Spectrum) at a wavelength of 450 nm with a reference reading of 630 nm. The optical density values from vehicle-control-treated cells with 0.2% DMSO were considered as 100% of cell viability. The experiments were triplicated to check the consistency. The LC_50_ values were computed using the GraphPad Prism 8.0.1 software by plotting a cell viability versus concentration graph. The percentage of cell viability was calculated using the formula below:% Cell viability = (Abs.Treated − Abs.Blank)/(Abs.Untreated − Abs.Blank) × 100%(1)

#### 4.2.2. Scratch Assay

The DBTRG-05MG were harvested and seeded in 6-well plates at 4 × 10^5^ cells/ml. Cell lines were cultured in 5% CO_2_ at 37 °C overnight to allow them to adhere to the plate. After cells reached confluency at 60–70%, the plate was scratched accordingly with 1 mL pipet tips, and the cell was washed once with PBS. Then, the cells were given treatment with an LC50 concentration of andrographolide (13.95 µM) and without andrographolide that was only treated with 0.2% of DMSO as vehicle control. The treated cells were incubated in 5% CO_2_ at 37°C for 24, 48, and 72 h. The following incubation cell was fixed and stained with 0.5% of crystal violet in 25% methanol for 1 h. The crystal violet was rinsed with tap water and dried on plates for a few days at room temperature. The plate was viewed and photographed on an inverted microscope at ×10 magnification. The wound area was evaluated using ImageJ software.

#### 4.2.3. Clonogenic Assay

The clonogenic assay was performed to evaluate the effectiveness of andrographolide toward DBTRG-05MG by observing colony formation after treatment. The DBTRG-05MG was harvested using trypsin-EDTA and seeded in 6-well plates with 4 × 10^5^ cell/mL of cell density for each well. Then, the cells were incubated in 5% CO_2_ at 37°C overnight to allow them to attach to the plate. The cells were treated with varying concentrations of andrographolide (13.95 µM and 27.21 µM) and without andrographolide that was only treated with 0.2% of DMSO as vehicle control. The treated cells were cultured in 5% CO_2_ at 37 °C for 72 h. After incubation, the old medium was removed and the cells were rinsed with PBS. The fixation solution (acetic acid/methanol 1:7 (*v*/*v*)) was added to the plate and incubated for 15 min at 25 °C (room temperature). Following the removal of the fixation solution, 1 mL of 0.5 percent crystal violet in 25% methanol was applied to the plates. The plates were placed for 1 h at 37 °C in a 5% CO_2_ incubator. The crystal violet was removed by immersing in tap water to rinse it off. The plates were allowed to air dry on a tablecloth at room temperature for a few days and were further observed, and images were recorded for the stained monolayer. The formation of the colony was counted and analyzed with ImageJ software. The plating efficiency (PE) and surviving fraction (SF) were calculated by using the formula below:PE = (Number of colonies formed)/(Number of cells seeded) × 100%(2)
SF = ((Number of colony formed after treatment)/(Number of cells seeded) × PE) × 100%(3)

#### 4.2.4. Cell Cycle and Apoptosis Analysis

The DBTRG-05MG were cultured in a T25 flask at a concentration of 20 × 10^4^ cells/mL. Then, the cells were placed in a 5% CO_2_ incubator at 37 °C overnight to allow them to attach to the flask. The cells were treated with varying concentrations of andrographolide corresponding to LC_50_ (13.95 µM)/2LC50 (27.21 µM) and without andrographolide that was only treated with 0.2% of DMSO as vehicle control. The treated cells were cultured in a 5% CO_2_ incubator at 37 °C for 72 h. Cell cycle arrests were assessed by PI/RNase staining buffer. The treated cells were harvested, fixed, and permeabilized with 70% ice-cold ethanol overnight. Then, the fixed cells were centrifuged to remove ethanol and washed once with ice-cold PBS. The cell pellet was resuspended in 0.5 mL of PI/RNase staining buffer and incubated for 15 min at room temperature. The cell cycle distribution was analyzed using BD FACS CantoTM Flow Cytometer (BD Biosciences, San Jose, CA, USA) using MoD fit software (Verity Software House, Topsham, ME, USA), while apoptosis assay was assessed using annexin V-FITC/PI double staining. Cell pellets were obtained after treatment were collected and re-suspended with 0.1 mL of 1× Annexin V Binding Buffer (AVBB), 5 µL of annexin V-FITC, and 5 µL of PI staining added in the tube. The cells were incubated in the dark at room temperature for 30 min. Flowjo software was used to analyze the stained cells using a BD FACS CantoTM Flow Cytometer (BD Biosciences, San Jose, CA, USA).

#### 4.2.5. Isolation RNA

Total RNA was extracted from the DBTRG-05MG cell line treated with varying concentrations of andrographolide correspondent to LC_50_ (13.95 µM)/2LC_50_ (27.21 µM) and without andrographolide that was only treated with 0.2% of DMSO as vehicle control. According to the official guidelines, the AurumTM Total RNA Mini Kit (Bio-Rad Laboratories, Inc, USA) was used to isolate total RNA from DBTRG-05MG cells. The NanoDrop 2000/2000c spectrophotometer was used to measure the RNA concentration (Thermo Fisher Scientific, Roskilde, Denmark). Electrophoresis on a 1% agarose gel was used to evaluate the integrity of total RNA. For use in qRT-PCR later, total RNA was kept at 80 °C.

#### 4.2.6. Quantitative Real-Time Reverse Transcription–Polymerase Chain Reaction (qRT-PCR)

The qRT-PCR was carried out using a Real-Time PCR machine, an iTaq Universal SYBR Green One-Step Kit on Qiagen Rotor-Gene Q-Pure detection (Corbett Research, Sydney, Australia). All reactions on each instrument were optimized to obtain the best amplification kinetics. The RNA template of each sample representing one biological triplicate was diluted to a working concentration of 30 ng/µL for the qRT-PCR analysis. The reaction setup and program thermal cycling protocol were followed according to the manufacturer’s protocol. A 5-point 10-fold serial dilution standard curve of RNA was used to generate a standard curve, while PCR efficiency (E) values ranging from 96.67 % to 109.04% were calculated over all primer sets with average correlations (R2) of 0.99. Primers used in this study were designed using NCBI/Primer-Blast and Primer3Plus as follows: GAPDH, FW 5’-ACATCGCTCAGACACCATG-3’ and RW 5’- TGTAGTTGAGGTCAATGAAGGG-3’; ERK1/2, FW 5’-CATTCAGCTAACGTTCTGCAC-3’ and RW 5’-GTGATCATGGTCTGGATCTGC-3’; c-Myc, FW 5’- TCCTCGGATTCTCTGCTCTC-3’ and RW 5’-TCTTCCTCATCTTCTTGTTCCTC-3’; p53, FW 5’- GACACGCTTCCCTGGATTG-3’ and RW 5’-GACGCTAGGATCTGACTGC-3’. The housekeeping glyceraldehyde-3-phosphate dehydrogenase (GAPDH) was used as a reference gene for comparing the relative expression of cDNA using the 2-△△CT method.

#### 4.2.7. Preparation of Total Cell Lysate

The total cell lysate was extracted from the DBTRG-05MG cell line treated with varying concentrations of andrographolide correspondent to LC_50_ (13.95 µM)/2LC_50_ (27.21 µM), along with cells treated with 0.2% of DMSO as vehicle control. The total cell lysate was isolated from DBTRG-05MG cells using the RIPA lysis buffer (Nacalai Tesque, Inc, Kyoto, Japan) according to the manufacturer’s instructions. The cells, both treated and untreated, were collected and washed twice in ice-cold phosphate-buffered saline (PBS). After, 1 mL 1× RIPA lysis buffer containing protease inhibitor cocktail was added to the cell pellet with occasional vortex. The lysed suspension was incubated for 30 min on ice and the following centrifugation at 10,000× *g* for 10 min at 4 °C was performed. The cell supernatant was transferred into new tubes immediately for further use in Western blot analysis and immunoprecipitation assays.

#### 4.2.8. Co-Immunoprecipitation

The protein G Sepharose beads (193259, Abcam) were activated by rinsing three times with 1000 µL of lysis buffer and centrifuging at 150× *g*. Total cell lysates obtained from the previous method were pre-cleared with 50 µL protein G Sepharose beads and incubated at 4 °C for 1 h, followed by gentle mixing using a rotating mixer. The pre-cleared lysate was recovered by centrifugation at 1200× *g* for 5 min and precipitated with the protein of interest antibody (Ab). The 5 µL of c-Myc Ab was added into pre-cleared lysate and incubated overnight at 4°C with moderate mixing on a rotating mixer. The immune protein complex was recovered by centrifugation at 1200× *g* for 5 min at 4 °C, then gently resuspended with cold lysis solution 3 times. The immune protein complex was recovered by centrifugation at 1200× *g* for 5 min at 4 °C, and the protein concentration was quantified by Bradford assay using a Bio-Rad protein assay kit. The immune protein complex was probed by Western blot analysis.

#### 4.2.9. Western Blot

The equal amount of denatured protein sample (30 µg) for each well was resolved on the 12% polyacrylamide gels and transferred to Bio Trace™ NT Nitrocellulose Transfer Membrane (PALL) by electroblotting. The membrane was blocked with 5% skim milk (Sun Lac) in 0.05% TBS-T for 1 h and stored at 4 °C. The membranes were then washed three times with TBST for five minutes each time before being incubated overnight at 4 °C with primary antibodies anti-p53 (1:500), anti-c-Myc (1:500), and anti-tubulin (1:500). After that, the membranes were probed with HRP-conjugated secondary antibodies for 2 h at 4 °C. The signal protein bands were detected using a Pierce^TM^ ECL Western Blotting Substrate (Thermo Fisher Scientific) and were developed in a dark room. The relative density of the interest protein band to the control protein (α-tubulin) was analyzed with ImageJ.

#### 4.2.10. Statistical Analysis

All the experiments were performed in triplicate, and the results were expressed as mean ± standard error of the mean (SEM). Other data were analyzed in GraphPad Prism version 8.0.1 using a two-way analysis of variance (ANOVA), proceeded by the Student’s t-test and Tukey’s test of one-way analysis. The *p* ≤ 0.05 was accepted as statistically significant.

## 5. Conclusions

This study successfully demonstrated the role andrographolide in inducing a cytotoxic effect in DBTRG-05MG cell lines in a time- and concentration-dependent manner. Further, andrographolide was found to reduce colony formation and delayed cell migration into the wound area. This study also evidenced that andrographolide induced a cytotoxic effect via G2/M phase arrest followed by apoptosis via the ERK1/2/c-Myc/p53 signaling pathway in a concentration-dependent manner. To our knowledge, this is the first work to show the mechanisms of andrographolide cell death in the DBTRG-05MG cell line. Because of its efficacy in inhibiting the DBTRG-05MG cell line, andrographolide also has the potential to be one of the anticancer treatments for glioblastoma multiforme disease. However, further study needs to be performed to evaluate the cytotoxicity of andrographolide in vivo and in vitro. The function of other regulatory genes that may interact with the c-Myc signaling pathway must also be investigated intensively.

## Figures and Tables

**Figure 1 molecules-27-06686-f001:**
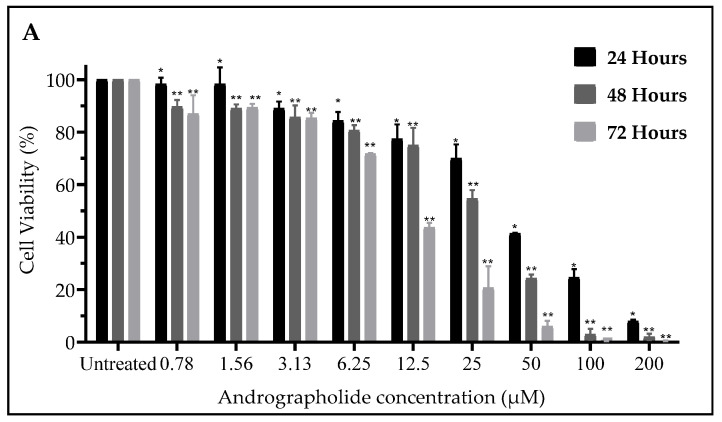
The percentage of cell viability of (**A**) DBTRG-05MG, (**B**) SVGp12 treated with andrographolide, (**C**) DBTRG-05MG treated with temozolomide in a time frame of 24, 48, and 72 h. The results were analyzed in GraphPad Prism 8.0.1 (Graphpad software, Inc., San Diego, CA, USA). Data expressed as mean ± SEM and statistical analysis between three independent groups was performed by t-test compared with the untreated group: * *p* < 0.05, ** *p* < 0.005, *** *p* < 0.0005.

**Figure 2 molecules-27-06686-f002:**
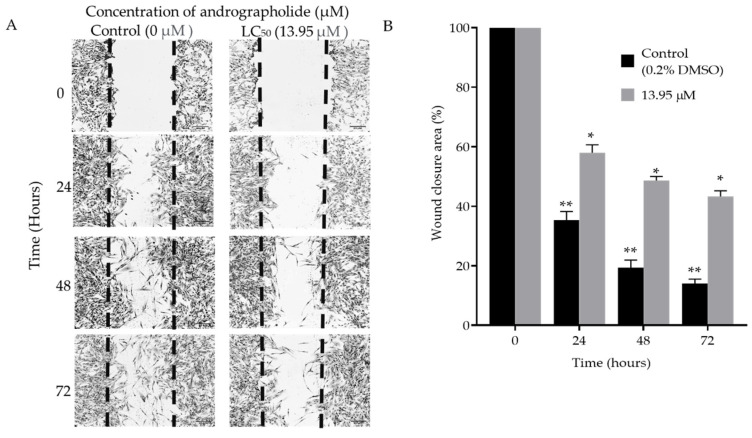
The effect of andrographolide in reducing migration of DBTRG-05MG cell lines. (**A**) The closure of the wound-healing area after 24, 48, and 72 h of DBTRG-05MG cell line treated with 13.95 μM andrographolide and without andrographolide as the control. (**B**) The % comparison of wound closure area of DBTRG-05MG cell lines when treated and untreated with andrographolide at 0, 24, 48, and 72 h. Data expressed as mean ± SEM and statistical analysis between three independent groups were performed by multiple t-tests compared with the control group (* *p* < 0.005, ** *p* < 0.0005).

**Figure 3 molecules-27-06686-f003:**
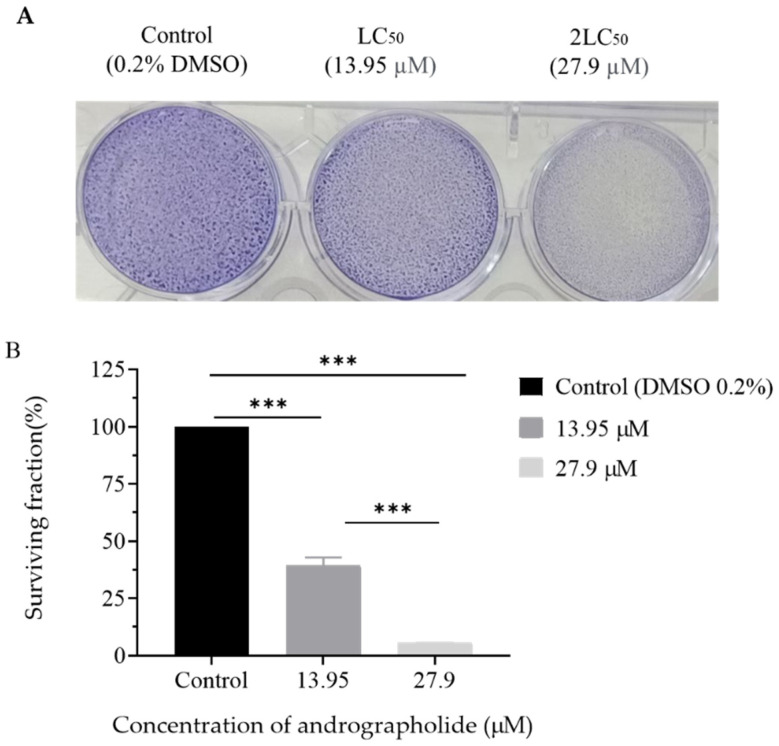
The effect of andrographolide in reducing colony formation of DBTRG-05MG cell lines. (**A**) Colony formation of DBTRG-05MG cells treated with 13.95 μM andrographolide and without andrographolide as the control. (**B**) The percentage of surviving fraction of DBTRG-05MG cell lines when treated with 13.95 μM and 27.9 μM of andrographolide compared with a control group. The comparison between each treatment with control was made using one-way ANOVA with Tukey’s test to detect any significant differences (*** *p* > 0.001).

**Figure 4 molecules-27-06686-f004:**
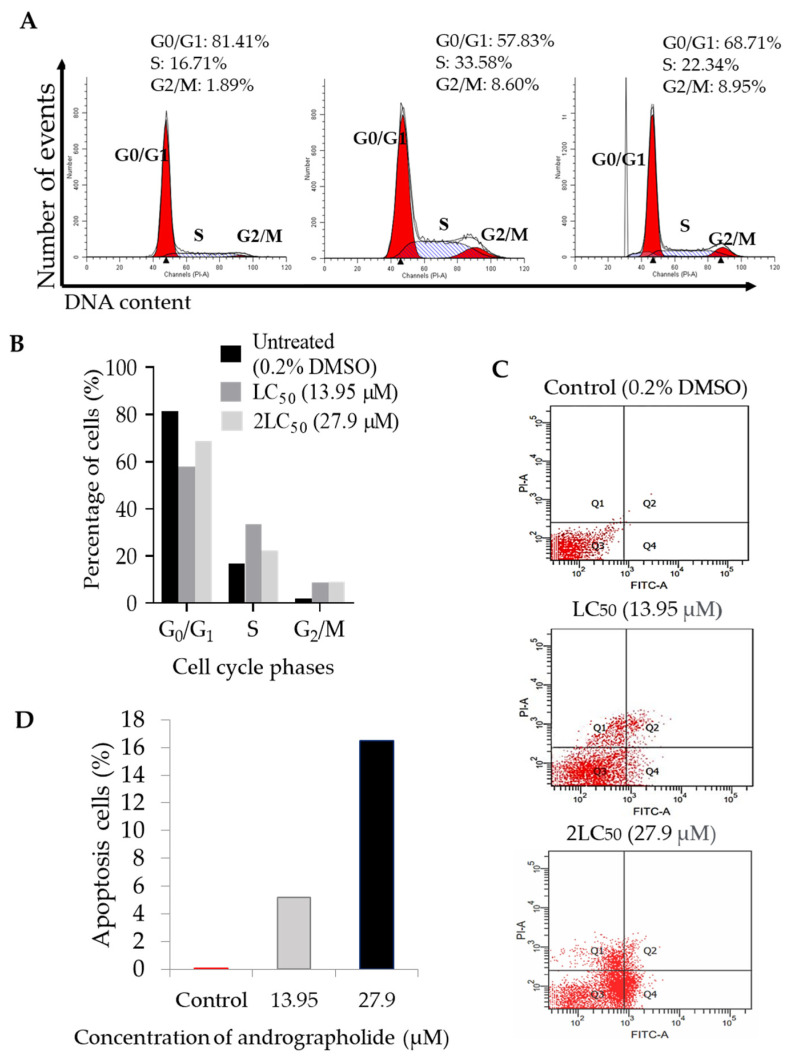
The cell cycle distribution of DBTRG-05MG cell lines. (**A**) The cell was distributed in the population of G0/G1, S, and G2/M of cell cycle phases after 72 h exposed to different concentrations of andrographolide (13.95 μM and 27.9 μM) compared with untreated cells (0.2% DMSO). (**B**) The percentage of cells (%) for each cell cycle phase is shown in the bar graph. (**C**) Apoptosis analysis of DBTRG-05MG cell lines after exposure with LC_50_ (13.95 μM), 2LC_50_ (27.9 μM) of andrographolide, and control (0.2% of DMSO) were observed using annexin V-FITC and PI staining. (**D**) The percentage of apoptosis cells increased after the concentration of andrographolide increased compared to control.

**Figure 5 molecules-27-06686-f005:**
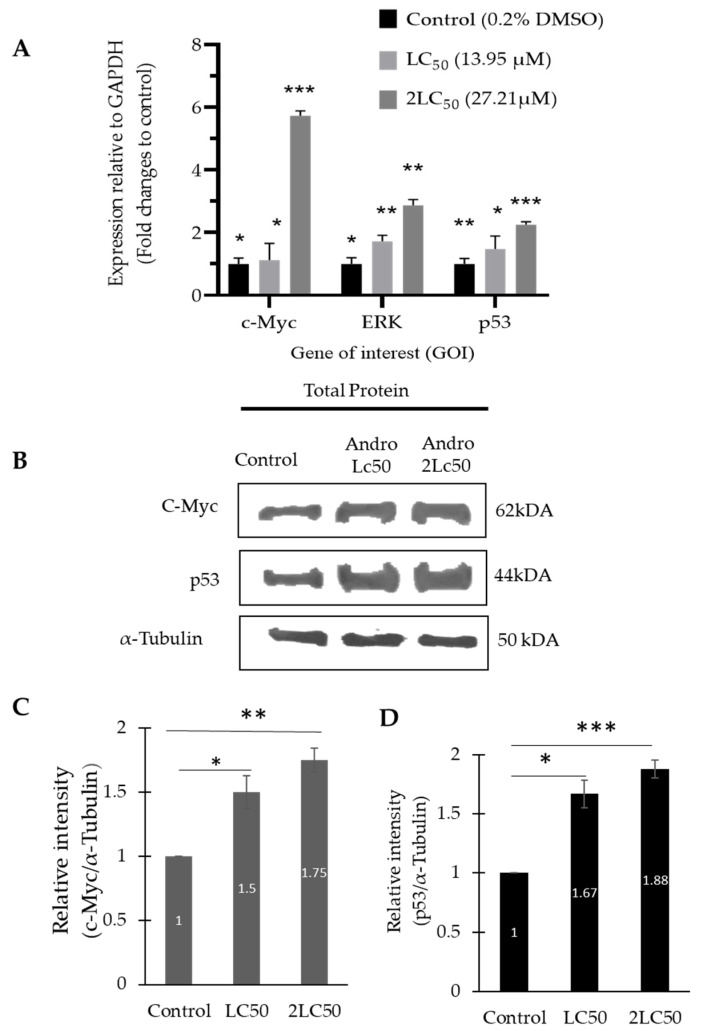
(**A**) Real-Time PCR (qPCR) results expressed relative changes of c-Myc, p53, and ERK over the control, using GAPDH as an endogenous housekeeping gene in DBTRG-05MG. (**B**) Representative Western blot analysis of p53 and c-Myc from DBTRG-05MG total cell lysate. The level of these protein expressions was normalized to α-tubulin. Band intensities were quantified using Image J software. Relative intensities of (**C**) c-Myc and (**D**) p53 are presented in the bar graph. Data expressed as mean ± SEM and statistical analysis between three independent groups was performed by t-test compared with the control group: * *p* < 0.05, ** *p* < 0.005, *** *p* < 0.0005.

**Figure 6 molecules-27-06686-f006:**
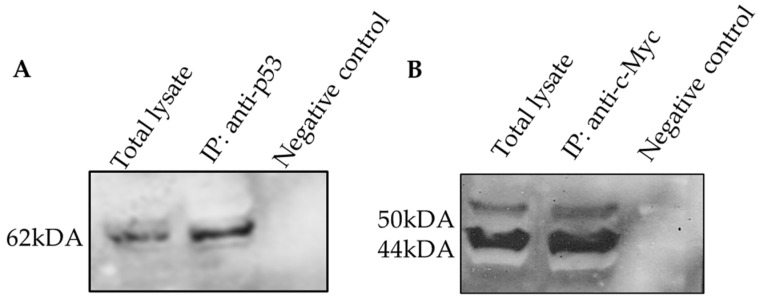
(**A**) Lane 1 shows the result of DBTRG-05MG total protein lysate probed with anti-C-Myc Ab; lane 2 shows the result of total proteins immunoprecipitated with anti-p53 and probed with anti-c-Myc in Western blot; and Lane 3 negative control. (**B**) Lane 1 is the result of DBTRG-05MG total protein lysate probed with anti-p53 antibodies; Lane 2 shows the result of total proteins immunoprecipitated with anti-c-Myc and probed with anti-p53 in Western blot.

**Figure 7 molecules-27-06686-f007:**
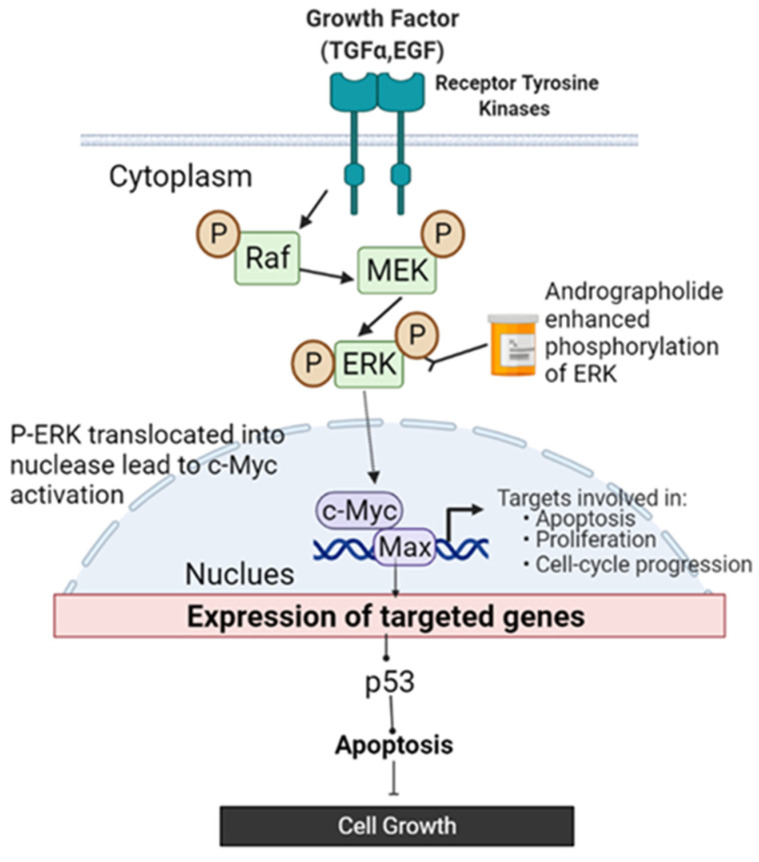
Schematic presentation of the proposed mechanism of the anticancer effect of andrographolide on DBTRG-05MG cells. The ERK/c-Myc/p53 signaling pathway was involved in andrographolide-induced apoptosis.

**Table 1 molecules-27-06686-t001:** The half-maximal lethal concentration (LC_50_) values for (**A**) DBTRG-05MG and SVGp12 cells after being treated with andrographolide (0.781–200 µM) and (**B**) DBTRG-05MG cells after being treated with temozolomide (0.781–200 µM) were estimated using nonlinear regression approximations in GraphPad Prism 8.0.1 (Graphpad software, Inc., San Diego, CA, USA).

**A**	**Cell Line**	**Treatment (h)/LC_50_ Values (µM)**		
		**24**	**48**	**72**
	DBTRG-05MG	42.82	27.21	13.93
	SVGp12	>200(n.d) 1	>200(n.d) 1	>200(n.d) 1
**B**	**Cell Line**	**Treatment (h)/LC_50_ Values (µM)**		
		**24**	**48**	**72**
	DBTRG-05MG	5.80	4.61	4.02

^1^ n.d. 1, not determined within the drug concentration range used.

## Data Availability

The data presented in this study are available on request from the corresponding author.

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
