# Peer review of "Andrographolide Induces G2/M Cell Cycle Arrest and Apoptosis in Human Glioblastoma DBTRG-05MG Cell Line via ERK1/2 /c-Myc/p53 Signaling Pathway"

_molecules, 2022, doi:10.3390/molecules27196686_

Round 1

Reviewer 1 Report

This research lacks novelty and originality. There are many published similar papers analysing the effect of andrographolide in glioblastoma cells, ERK1/2-related pathways, migration, etc.

Moreover, the concentrations of this substance used in biological experiments are equal to EC50 values, thus it means that the effects on cell migration and clonogenicity were determined mostly by cytotoxicity on cells – just part of the cells died due to toxicity and observed a lower number of cells in wound healing assay and clonogenic assay could be mistakenly related to some other specific effects. Also, it is not correct to make conclusions from only one cancer cell line, usually, biologists test the effects on at least two-three cell lines to avoid accidental effects on one cell line. Due to all mentioned reasons, I did not find the conclusions convincing.

Author Response

Dear Ms. Jelena Ratkovic and reviewer 1,

Thank you for allowing us to submit a revised draft of our manuscript titled Andrographolide induces G2/M cell cycle arrest and apoptosis in human glioblastoma DBTRG-05MG cell line via ERK1/2 /c-Myc/p53 signaling pathway to Molecules, MDPI journal. We appreciate the time and effort that you and the reviewers have dedicated to providing your valuable feedback on my manuscript. We are grateful to the reviewers for their insightful comments on our paper. We have been able to incorporate changes to reflect most of the suggestions provided by the reviewers. We have highlighted the changes within the manuscript.

Here is a point-by-point response to the reviewers’ comments and concerns.

Reviewer 1:

Comments and Suggestions from reviewer

Response to reviewer comments

Comment 1:

This research lacks novelty and originality. There are many published similar papers analyzing the effect of andrographolide in glioblastoma cells, ERK1/2-related pathways, migration, etc.

Thank you for pointing this out. While we appreciate your feedback, we respectfully disagree. We think this study has its novelty and originality because out of 110 research articles that studies investigated andrographolide there is only 3.16% of the study effect of andrographolide on brain disease, especially in GBM. The previous study already demonstrated that ERK1/2 pathways were related to andrographolide in GBM cells such as in GBM8401, U251, and U87 cell lines but not in DBTRG-05MG cell lines. Since each cell line is derived from the same disease but with different variations in phenotype and genotype from which the patient is suffering, it offers the opportunity for disclosing pathological features that were otherwise unidentified by conventional clinical diagnostic settings.  Then, our study also has similar findings related to ERK1/2 pathways but no study information related to ERK1/2/c-Myc/p53 pathways. However, in the case of our study was tried to highlight upstream and downstream c-Myc signaling pathways involved in the DBTRG-05MG cell line.

Comment 2:

Moreover, the concentrations of this substance used in biological experiments are equal to EC50 values, thus it means that the effects on cell migration and clonogenicity were determined mostly by cytotoxicity on cells – just part of the cells died due to toxicity and observed a lower number of cells in wound healing assay and clonogenic assay could be mistakenly related to some other specific effects.

Thank you for your comment and suggestion. The EC50 values are important to predict the toxicity of andrographolide in the DBTRG-05MG cell line and one of the optimization steps to choose the safe concentration of toxicant that can use in the next experiment. Then, cell migration and clonogenicity were observed in cell proliferation or growth after being induced with EC50 values of andrographolide.

Comment 3:

Also, it is not correct to make conclusions from only one cancer cell line, usually, biologists test the effects on at least two-three cell lines to avoid accidental effects on one cell line. Due to all mentioned reasons, I did not find the conclusions convincing.

You have raised an important point here. Although we agree that this is an important consideration. We believe it would have been interesting to explore different cell lines. However, our study tries to provide new information regarding the DBTRG-05MG cell line because the previous study already covered GBM8401, U251, and U87 glioblastoma multiforme cell lines but not in the DBTRG-05MG cell line. since each cell line is derived from the same disease but with different variations in phenotype and genotype. Maybe in future works, we tried to use a variety of GBM cell lines.

Additional clarifications:

In addition to the above comments, all spelling and grammatical errors pointed out by the reviewer have been corrected. We look forward to hearing from you in due time regarding our submission and to responding to any further questions and comments you may have.

Sincerely,

Daruliza

Dr. Daruliza Kernain Mohd Azman

Reviewer 2 Report

In the present study entitled "Andrographolide induces G2/M cell cycle arrest and apoptosis in human glioblastoma DBTRG-05MG cell line by via ERK1/2 /c-Myc/p53 signaling pathway" the antitumor potential of andrographolide in molecular pathways was evaluated using a DBTRG-05MG cell line.

The study carried out shows promising results in the perspective of using this compound as a therapeutic agent against cancer. The text is properly written, but it is necessary to make the corrections suggested below. Furthermore, I am in favor of accepting the manuscript for publication in the journal:

1)    In the Abstract remove the word natural from the term “natural plants”. All plants are natural, so it is not common to use this adjective. You can use another term, for example plant species.

2)    Remove the term sp. in the name of the plant, “Andrographis paniculata sp”. It is a mistake. The correct is “Andrographis paniculata

3)    The first time you write the name of the plant in the manuscript, you must enter the name of the botanist who classified the plant. For example “Andrographis paniculata (Burm.f.) Wallich ex Nees.”.

4)    In lines 54-55, the authors wrote "Andrographolide is a colorless crystalline solid with a bitter taste, and diterpenoid lactone is the major phytochemical found in it". Please correct this sentence. Andrographolide is the lactone diterpenoid. So the substance cannot be found in the substance itself.

5)    In lines 57-59 it says "andrographolide is a potential drug candidate for the treatment of any diseases related to the central nervous system (CNS) because it is a polar compound with low molecular weight [5] that is in crossing the blood-brain barrier". Why is the polarity of this compound related to its activity in the CNS? It is known that less polar compounds cross the blood-brain barrier more easily. Then clarify the sentence.

6)    I suggest inserting the chemical structure of andrographolide in the Introduction.

7)    In Conclusions, it is written "This study demonstrated the andrographolide’s potential to induce a cytotoxicity effect in DBTRG-05MG cell lines". I think the word 'potential' is not necessary because cytotoxic activity has been demonstrated. Please improve this sentence.

Author Response

Dear Ms. Jelena Ratkovic and reviewer 2,

Thank you for allowing us to submit a revised draft of our manuscript titled Andrographolide induces G2/M cell cycle arrest and apoptosis in human glioblastoma DBTRG-05MG cell line via ERK1/2 /c-Myc/p53 signaling pathway to Molecules, MDPI journal. We appreciate the time and effort that you and the reviewers have dedicated to providing your valuable feedback on my manuscript. We are grateful to the reviewers for their insightful comments on our paper. We have been able to incorporate changes to reflect most of the suggestions provided by the reviewers. We have highlighted the changes within the manuscript.

Here is a point-by-point response to the reviewers’ comments and concerns.

Reviewer 2:

Comments and Suggestions from reviewer

Response to a reviewer comment

Comment 1:

In the Abstract remove the word natural from the term “natural plants”. All plants are natural, so it is not common to use this adjective. You can use another term, for example, plant species

We think this is an excellent suggestion. The word “natural plants” has been corrected to plant species. We have added the suggested words to the manuscript on line 14

Comment 2:

Remove the term sp. in the name of the plant, “Andrographis paniculata sp”. It is a mistake. The correct is “Andrographis paniculata

Thank you for pointing this out. We have removed the term sp. in the Andrographis paniculata sp on lines 18 and 52

Comment 3:

The first time you write the plant's name in the manuscript, you must enter the name of the botanist who classified the plant. For example “Andrographis paniculata (Burm.f.) Wallich ex Nees.”.

We agree with this, and we have accordingly revised the name of the plant on line 18 with the Andrographis paniculata (Burm.f.) Wallich ex Nees

Comment 4:

In lines 54-55, the authors wrote "Andrographolide is a colorless crystalline solid with a bitter taste, and diterpenoid lactone is the major phytochemical found in it". Please correct this sentence. Andrographolide is the lactone diterpenoid. So the substance cannot be found in the substance itself.

As suggested by the reviewer, we have accordingly revised lines 54-56 and have updated to Andrographolide as the lactone diterpenoid with a colorless crystalline solid with a bitter taste.

Comment 5:

In lines 57-59 it says "andrographolide is a potential drug candidate for the treatment of any diseases related to the central nervous system (CNS) because it is a polar compound with low molecular weight [5] that is in crossing the blood-brain barrier". Why is the polarity of this compound related to its activity in the CNS? It is known that less polar compounds cross the blood-brain barrier more easily. Then clarify the sentence.

Thank you for pointing this out. We agree with this comment. As a response, we clarified the sentence on lines 62-67 in the amended document.

Comment 6:

I suggest inserting the chemical structure of andrographolide in the Introduction.

We agree with this and have incorporated your suggestion throughout the manuscript in lines 58-60.

Comment 7:

In Conclusion, it is written, "This study demonstrated the andrographolide’s potential to induce a cytotoxicity effect in DBTRG-05MG cell lines". I think the word 'potential' is not necessary because the cytotoxic activity has been demonstrated. Please improve this sentence.

Thank you for this suggestion. We have accordingly changed this sentence, “This study demonstrated the andrographolide’s potential to induce a cytotoxicity effect in DBTRG-05MG cell lines" into “This study successfully demonstrated the andrographolide in inducing a cytotoxicity effect in DBTRG-05MG cell lines" (line 567-568).

Additional clarifications:

In addition to the above comments, all spelling and grammatical errors pointed out by the reviewer have been corrected. We look forward to hearing from you in due time regarding our submission and to responding to any further questions and comments you may have.

Sincerely,

Daruliza

Dr. Daruliza Kernain Mohd Azman

Reviewer 3 Report

Congratulations for the work. It is a very interesting paper and talk about important topics on neurological disorders and GBM. I just suggest a minor grammatical review.

Author Response

Dear Ms. Jelena Ratkovic and reviewer 3,

Thank you for allowing us to submit a revised draft of our manuscript titled Andrographolide induces G2/M cell cycle arrest and apoptosis in human glioblastoma DBTRG-05MG cell line via ERK1/2 /c-Myc/p53 signaling pathway to Molecules, MDPI journal. We appreciate the time and effort that you and the reviewers have dedicated to providing your valuable feedback on my manuscript. We are grateful to the reviewers for their insightful comments on our paper. We have been able to incorporate changes to reflect most of the suggestions provided by the reviewers. We have highlighted the changes within the manuscript.

Here is a point-by-point response to the reviewers’ comments and concerns.

Reviewer 3:

Comments and suggestions from reviewer

Response to a reviewer comment

Congratulations on the work. It is a very interesting paper and talks about important topics on neurological disorders and GBM. I just suggest a minor grammatical review.

We appreciate your valuable feedback and efforts to improve our manuscript. All of the grammatical errors have already been corrected.

Additional clarifications:

In addition to the above comments, all spelling and grammatical errors pointed out by the reviewer have been corrected. We look forward to hearing from you in due time regarding our submission and to responding to any further questions and comments you may have.

Sincerely,

Daruliza

Dr. Daruliza Kernain Mohd Azman

Round 2

Reviewer 1 Report

Sorry, but in my opinion the design of this research is not correct, as the toxic concentrations were used for experiments. For example, for migration assay. It is not correct at all state then that the cells were not migrating, as they actually were most probably dying... Also, one cell line is not enough for studies, at least two different cancer cell lines are used for such studies, as it is according to the guidelines of most highly-rated journals, to avoid accidental findings